# Plant Performance and Soil Microbial Responses to Irrigation Management: A Novel Study in a Calafate Orchard

**Matías Betancur [1], Jorge Retamal-Salgado [2,\*], María Dolores López [3], Rosa Vergara-Retamales [4] and Mauricio Schoebitz [1,5,\*]**

1    Departamento de Suelos y Recursos Naturales, Facultad de Agronomía, Universidad de Concepción, Concepción 4030000, Chile

2    Instituto de Investigaciones Agropecuarias, INIA-Quilamapu, Avenida Vicente Méndez 515, Chillán 3800062, Chile

3    Departamento de Producción Vegetal, Facultad de Agronomía, Universidad de Concepción, Chillán 3780000, Chile

4    Faculty of Engineering and Business, Universidad Adventista de Chile, Km 12 Camino a Tanilvoro, Chillán 3780000, Chile

5    Center of Biotechnology, Laboratory of Biofilms and Environmental Microbiology, Universidad de Concepción, Barrio Universitario s/n, Concepción 4030000, Chile

*    Correspondence: jorge.retamal@inia.cl (J.R.-S.); mschoebitz@udec.cl (M.S.)

**Abstract:** Calafate fruits have a high content of phenolic compounds and an antioxidant activity up to four times higher than that of blueberries. The establishment of a calafate orchard and irrigation responses on fruit and soil characteristics have been scarcely studied. Therefore, the objective of this study was to evaluate the effect of water replenishment rate: 0%, 50%, 100% and 150% of reference evapotranspiration ($ET_0$), on soil microbiological activity, plant physiological response, fruit yield and chemical composition in a calafate orchard. The results showed that irrigation at 50% $ET_0$ presented significant increases in soil urease, dehydrogenase and acid phosphatase activity. Likewise, irrigation at 50% $ET_0$ significantly increased stomatal conductance and plant chlorophyll index, which led to a significant increase in fruit yield being 60% higher compared to the other treatments. Despite the higher fruit yield, 50% $ET_0$ irrigation had a similar level of total anthocyanins and ORAC antioxidant capacity as the 100% $ET_0$ treatment. In contrast, 0% and 150% $ET_0$ treatments showed a higher degree of stress and got higher values for total anthocyanins and fruit antioxidant capacity. Irrigation rates 50% $ET_0$ increases fruit yield while maintaining fruit quality and optimizing water resources in commercial orchards of calafate.

**Keywords:** berberis; soil enzyme activity; polyphenols; anthocyanins; antioxidant capacity

## 1. Introduction

*Berberis microphylla* G. Forst., commonly called calafate, is a very thorny shrub-like species, with arching branches and blackish-blue berries, which grows under different agroclimatic conditions in Chile and Argentina [1].

Currently, the calafate has become known for the properties of its blackish blue fruits, as they have a high content of phenolic compounds and antioxidant activity [2], being up to ten times higher than oranges, apples and pears, and up to four times more than blueberries [3]. Despite its functional and nutritional properties, it is only grown in the wild, without cultural management to improve its phytosanitary status, and longevity of branches or shoots [4], which leads to a low fruit production per plant, resulting in total production of calafate fruit in Chile that did not exceed 1 t in 2019 [5]. In this context, to stimulate its productive development, in 2017, the first commercial orchard was established in the central-southern zone of Chile, for domestication and development of agronomic management [6]. Pinto-Morales et al. [6] pointed out that agronomic management is being

addressed, but the implementation of commercial orchards brings new questions, such as how the interaction of these agronomic managements is altering soil biological properties.

The calafate is a highly adaptable species, being able to develop and bear fruit in regions with a Mediterranean climate [7]. However, it has been shown that environmental factors such as ultraviolet (UV) radiation and temperature can alter phenol and anthocyanin content of the fruit [8], a relevant factor due to the high impact of environmental changes resulting from climate change reported in recent times [9] and the current food trend, which demands healthier foods [10]. Therefore, the evaluation of water replenishment for the establishment of this species becomes relevant, since this management practice attenuates soil water losses [11] and defines nutritional status of the plant, which affects fruits production and chemical characteristics [12]. In this sense, water replacement in fruit crops is used as a strategy to enhance the accumulation of secondary metabolites in the fruit [13] and to reduce the consumption and losses of water, which is increasingly scarce [14]. Accordingly, it has been shown that the use of controlled deficit irrigation (CDI) benefited 'Barnea' and 'Askal' olive oil quality, reaching mean polyphenol contents of up to 372 and 487 mg kg$^{-1}$, respectively, which represented increases of up to 150% and 385%. However, it was also shown that this type of controlled water stress produced lower fruit yields, from 305 to 265 kg/tree$^{-1}$ in 'Barnea' and from 205 to 190 kg/tree$^{-1}$ in 'Askal', respectively [14]. Consequently, knowing the water requirements of native plants of high nutritional and pharmacological value [2] would allow for setting up the best establishment conditions to increase the quality of fruits and efficiency of water resources.

It should be noted that variation in water availability alters soil physical [15], chemical [16] and microbiological properties [17]. The latter is relevant because, according to several authors, some soil bacterial communities can alter the functional characteristics of plants [18] and allow greater resistance to pathogen invasion [19]. Likewise, soil fungi such as arbuscular mycorrhizal fungi (AMF) form symbiotic associations with roots and improve the hormonal balance of the plant to resist water deficits [20,21]. In accordance with the above, previous studies have shown that soil water availability through different doses can determine the size, distribution, and activity of soil microorganisms [22], with bacterial communities being more sensitive to changes in soil moisture and osmotic stress than fungi [23], which is consistent with what has been reported in an almond orchard under different water management strategies [24].

To date, there are no studies that evaluate the interaction of soil microorganisms with the degree of adaptability and chemical characteristics of the fruit, nor how these would be influenced by different levels of water replenishment. In fact, as far as fruit-bearing species are concerned, studies showing the influence of irrigation dose on soil microbiological properties and plant productive parameters are still incipient and their results are dissimilar. For example, in a grapefruit orchard irrigated with different water quantities, restricted water replenishment decreased soil microbial biomass by up to 33%, leading to a decrease in fruit production from 232 to 184 kg/fruit tree$^{-1}$ [24]. In contrast, in an almond orchard, restricted water replenishment significantly increased soil microbial biomass by up to 131%, as well as the activity of enzymes such as urease by 148% and phosphatase by 102%, but reduced fruit production by 86% [25].

This is the first study that proposes to evaluate the water replenishment rates of calafate under a commercial system and its effect on soil microbiological and enzyme activity, plant physiology, fruit production, and fruit secondary metabolite accumulation.

## 2. Materials and Methods

### 2.1. Orchard Establishment and Edaphoclimatic Characteristics of the Study Site

The calafate (*Berberis microphylla* G. Forst.) orchard was established in 2017 with two-year-old plants. The orchard was located on the road to Tanilvoro, kilometer 12, from Chillán (36°31′ S; 71°54′ W), Ñuble Region, Chile, with a temperate Mediterranean climate with mean annual temperatures of 14.5 °C and maximum temperatures of 31.1 °C in the hottest months, and accumulated annual precipitation of 521.1 mm concentrated between

May and October [26]. The Andisol (Melanoxerand) [27] was characterized by chemical analysis at the soil laboratory of the Instituto de Investigaciones Agropecuarias (INIA Quilamapu), located in Chillan (Table 1).

**Table 1.** Soil chemical analysis of the study site in the Ñuble Region of Chile.

| Analysis | Unit | Result |
|---|---|---|
| Organic matter | % | 9.7 |
| pH (water) | | 6.4 |
| N availability | mg kg$^{-1}$ | 19.0 |
| Olsen P | mg kg$^{-1}$ | 15.3 |
| K availability | mg kg$^{-1}$ | 496.0 |
| S availability | mg kg$^{-1}$ | 24.0 |
| Exchangeable Ca | cmol$_+$ kg$^{-1}$ | 8.7 |
| Exchangeable Mg | cmol$_+$ kg$^{-1}$ | 1.6 |
| Exchangeable K | cmol$_+$ kg$^{-1}$ | 1.3 |
| Exchangeable Na | cmol$_+$ kg$^{-1}$ | 0.01 |
| Sum of bases | cmol$_+$ kg$^{-1}$ | 11.6 |
| Interchangeable Al | cmol$_+$ kg$^{-1}$ | 0.02 |
| CEC * | cmol$_+$ kg$^{-1}$ | 11.6 |
| Al saturation | % | 0.1 |
| B | mg kg$^{-1}$ | 0.4 |
| Cu | mg kg$^{-1}$ | 1.6 |
| Zn | mg kg$^{-1}$ | 0.9 |
| Fe | mg kg$^{-1}$ | 44.0 |
| Mn | mg kg$^{-1}$ | 3.0 |

* CEC: Cation exchange capacity of the soil; samples were obtained in August 2020, at a depth of 0–40 cm.

The total number of plants in the orchard was 352 in an area of 1056 m$^2$, distributed in 16 plants per row with a total of 22 rows and a planting density of 1 m above row and 3 m between rows (Figure 1a,b). Only at the time of orchard establishment, fertilization was applied with 150 g urea (45% N), 200 g triple superphosphate (46% P$_2$O$_5$), and 200 g potassium sulfate (50% K$_2$O) per planting hole [28]. Phytosanitary management consisted of six alternating applications per year of tebuconazole (Orius 43 SC, Adama Brasil, Londrina B, Brasil) at a concentration of 25.8 g hL$^{-1}$ and cuprous oxide (Cuprodul WG, Quimetal Industrial S.A., Santiago, Chile) at a concentration of 180 g hL$^{-1}$, during the first year of orchard establishment.

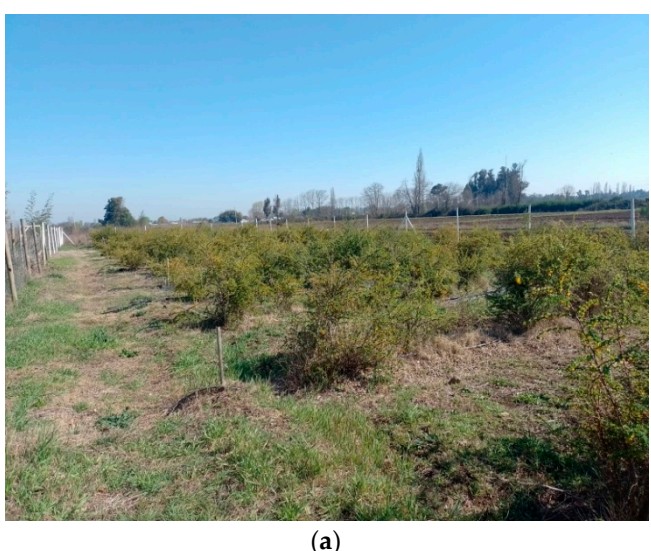
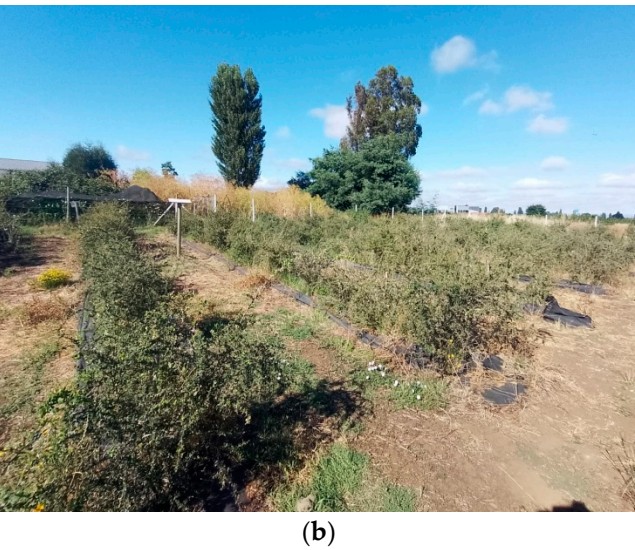

(**a**) (**b**)

**Figure 1.** (**a**) Flowering calafate orchard, September 2021. (**b**) calafate orchard in fruiting, December 2021.

## 2.2. Experimental Setup of the Test

Four irrigation treatments were established consisting of 0%, 50%, 100%, and 150% water replenishment of reference evapotranspiration ($ET_0$). The irrigation treatments were applied from the establishment of the orchard (year 2017) in the same period indicated for the evaluation. To determine the actual daily irrigation, $ET_0$ was calculated monthly using the Pen-man-Monteith method, taking as a reference what was suggested by Romero et al. [28], with data obtained from the INIA Quilamapu agroclimatic station located near the study site [29]. Irrigation treatments were applied using one irrigation lateral for the 50% ET0 treatment, two irrigation laterals for the 100% $ET_0$ treatment, and three irrigation laterals for the 150% $ET_0$ treatment, with self-compensating pressure drippers (UniRam, Netafim, Hatzerim, Israel) spaced at 50 cm with a flow rate of 2.0 $Lh^{-1}$. The control treatment did not consider irrigation laterals. The irrigation period for the evaluation started in September 2020, until March 2021 (Figure 2), with a total evapotranspiration of 828.5 mm. The actual daily irrigation was applied every two days, where the daily irrigation time (h $day^{-1}$) was calculated based on the monthly evapotranspiration (mm) divided by the number of ir-rigation days, being the actual monthly evapotranspiration and daily irrigation time for September were 33 mm and 2.2 h; October 53 mm and 3.5 h; November 65 mm and 4.3 h; December 166 mm and 5.5 h; January 160 mm and 5.3 h; February 107 mm and 3.6 h; and March 92 mm and 3 h, respectively.

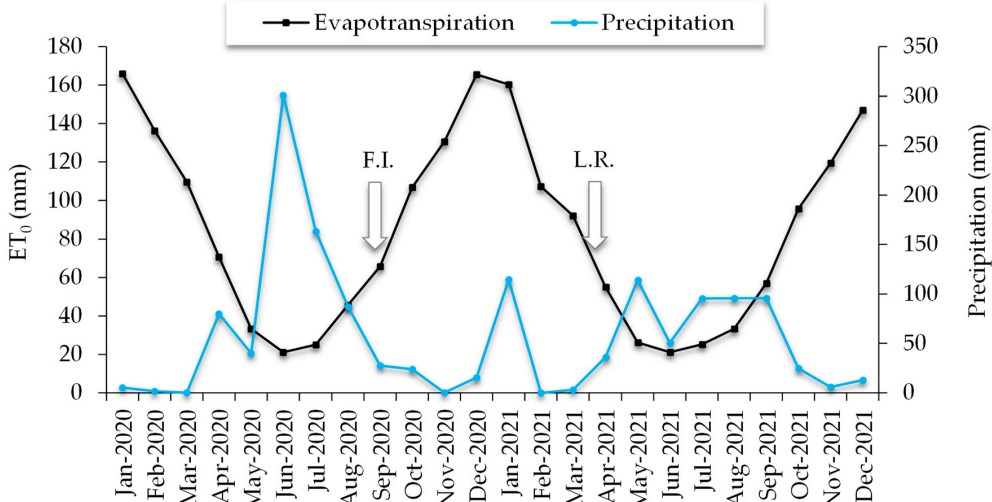

**Figure 2.** Irrigation period for calafate plants: FI: First irrigation; LR: last watering, based on reference evapotranspiration ($ET_0$) and precipitation, according to information from the agroclimatic station of the Instituto de Investigaciones Agropecuarias (INIA Quilamapu), Chile.

Soil volumetric water content (%) was monitored every 15 days (Table 2) with a Diviner 2000 portable soil moisture probe (Sentek, Stepney, Australia) that provides the soil moisture value in percent, from November 2021 to February 2022. The statistical design was randomized complete blocks. Soil microbiological, plant physiological and fruit chemical analyses were determined only once at the end of the harvest period, during the month of January. Samples were taken from each experimental unit, where the experimental unit consisted of the average of two subsamples, each subsample being a calafate plant, and four replicates per treatment (*n* = 16) were carried out in all blocks.

**Table 2.** Soil moisture monitoring with a frequency of 15 days.

| Treatments ($ET_0$ %) | Soil Moisture (%) | | | | | | | |
|---|---|---|---|---|---|---|---|---|
| | 15-November-2021 | 30-November-2021 | 15-December-2021 | 30-December-2021 | 15-January-2022 | 30-January-2022 | 15-February-2022 | 30-February-2022 |
| 0 | 6.54 | 5.53 | 3.12 | 2.30 | 6.18 | 5.79 | 2.00 | 1.93 |
| 50 | 9.68 | 11.43 | 7.70 | 5.43 | 21.73 | 28.66 | 24.43 | 8.80 |
| 100 | 10.71 | 13.25 | 9.25 | 7.23 | 24.85 | 31.77 | 30.63 | 9.60 |
| 150 | 15.57 | 16.55 | 11.40 | 8.80 | 30.02 | 35.10 | 34.90 | 10.65 |

Treatments: 0%, 50%, 100% and 150% $ET_0$. Measurements on 15 November 2021, 30 November 2021, 15 December 2021, 30 December 2021, 30 December 2021, and 30 February 2022 were measured 48 h after irrigation. Measurements on 15 January 2022, 30 January 2022, and 15 February 2022 were measured 24 h after irrigation.

*2.3. Microbiological Analysis of Soil and Roots*

Soil microbiological activity was determined by fluorescein diacetate (FDA) hydrolysis using 1.0 g wet soil sample in triplicate including a blank. In glass tubes, 9.9 mL sodium phosphate buffer (60 mM; pH 7.8) and 0.1 mL FDA were added, while only 10 mL buffer were added to the blanks, then vortexed and incubated at 20 °C for 1 h in a thermostatic bath. After incubation, the reaction was stopped in an ice water bath and 10 mL acetone was added to each tube, stirring to homogenize, and then filtered using Whatman No. 40 filter paper. Once each sample was obtained, the absorbance was measured using a spectrophotometer (Rayleigh-Model UV1601 UV/VIS, Beijing, China) at 490 nm. The results were expressed as $\mu$g FDA g$^{-1}$ [30].

Soil respiration was determined in duplicate using 25 g soil, which was placed in a flask together with a centrifuge tube with a volume of 7.5 mL 0.5 M NaOH. Both were placed in a hermetically sealed system and incubated at 22 °C for 7 days. After the incubation time, 1 mL 0.5 M NaOH was taken from the centrifuge tube and mixed with 2 mL 1 M $BaCl_2$; 2 to 3 drops of phenolphthalein were previously added as an indicator. The solution was titrated with 0.1 M HCl and data were expressed as $\mu$g $CO_2$ g$^{-1}$ h$^{-1}$ [30].

Urease activity was determined in 0.1 M phosphate buffer at pH 7 using 1 M urea as substrate. Aliquots of 2 mL buffer and 0.5 mL substrate were added to 0.5 g sample and incubated at 30 °C for 90 min. Urease activity was expressed as $NH_4^+$ released in the hydrolysis reaction [31], according to the ammonium sulfate standard curve. The standard curve was prepared with a dilution of 4 mL of ammonium sulfate in 200 mL of distilled water. Amounts of 0, 2, 4, 6, 6, 8, 10 and 12 mL of the dilution were added in 25 mL volumetric flasks to which 5 mL of EDTA, 1 mL of phenol nitroprusside, and 4 mL hypochlorite buffer were added. The absorbance was read against reagent blank at 636 nm. Once the data was recorded, it was fitted to a straight line of $\mu$g $NH4^+$ versus absorbance and $\mu$mol $NH4^+$ g$^{-1}$ h$^{-1}$ = ($NH4^+$ $\mu$g/mL)/dry soil weight (g)*Incubation time (h). Acid phosphatase activity was determined with *P*-nitrophenyl disodium phosphate (PNPP 0.115 M) substrate, for which, 2 mL 0.5 M sodium acetate buffer at pH 6 with acetic acid [32] and 0.5 mL substrate was added to 0.5 g soil sieved at <2 mm and then incubated at 37 °C for 90 min. The reaction was stopped by cooling to 0 °C for 10 min in a cuvette of water with crushed ice. Then, 0.5 mL 0.5 M $CaCl_2$ and 2 mL 0.5 M NaOH were added and the mixture was centrifuged at 1382 g. The *P*-nitrophenol (PNP) formed was determined spectrophotometrically at 398 nm [33]. Los $\mu$mol PNF g$^{-1}$ soil h$^{-1}$ = ((80.093*absorbance −0.4026)/peso*weight*incubation time*139).

Dehydrogenase activity was determined with 1 g soil at 60% field capacity exposed to 0.2 mL 0.4% INT (2-p-iodophenyl-3-p-nitrophenyl-5-phenyltetrazolium chloride) and carried in darkness for 20 h at 22 °C. The iodonitrotetrazoliumformazan (INTF) formed was extracted with 10 mL methanol by shaking for 1 min with a vortex and filtering through Whatman No. 5 filter paper. Subsequently, INTF was measured spectrophotometrically at 490 nm and the results were expressed in terms of micrograms of INTF per gram of soil

concerning an INTF standard curve [34], mg INTF $g^{-1}$ = ((39.997*absorbance-0.3834)*final extraction volume/weight).

The colonization of arbuscular mycorrhizal fungi (AMF) in roots was determined by cutting 1 cm pieces of fine roots which were washed and clarified with KOH (2.5% $w/v$) at 120 °C for 15 min. Subsequently, roots were covered with HCl (1% $w/w$) for 1 day and then washed with abundant water to cover them with trypan blue (0.05% $w/v$) for 1 day [35]. To determine the percentage of mycorrhizal colonization in the roots, these were randomly distributed on a grid plate to subsequently visualize structures such as mycelium, spores, hyphae, arbuscules, and vesicles in the root tissues through a microscope at 40× (Siedentopf, United Scope, Irvine, CA, USA) and by counting all root intersections with horizontal lines according to the presence or absence of AMF structures [36,37].

### 2.4. Plant Physiological Analysis

Maximum chlorophyll fluorescence ($F_m$) and minimum chlorophyll fluorescence ($F_o$) were measured in leaves on a clear day after harvest at four times of the day, at 09:00, 12:00, 15:00, and 18:00 h using a portable fluorimeter model OS-5p (Opti-Sciences, Hudson, NH, USA). For determination of $F_o$ and $F_m$, leaves were dark adapted using leaf clips including a movable shutter plate for 30 min [38]. Maximum photosystem II photochemical efficiency ($F_v/F_m$) was quantified using the ratio $F_v/F_m = (F_m - F_o)/F_m$ [39].

Stomatal conductance ($g_s$, mmol $m^{-2}$ $s^{-1}$) was measured on leaves on a clear day after harvest at four times of the day, at 09:00, 12:00, 15:00, and 18:00 h using a portable porometer model SC-1 (Decagon Devices, Washington, DC, USA). For data representativeness, the leaf selection criteria were leaves exposed to the sun, in the second third of a branch of the season as with the data obtained from chlorophyll fluorescence [38].

The SPAD (Soil Plant Analysis Development) index was measured on leaves at midday with the criteria used for chlorophyll fluorescence and $g_s$ measurements, after harvest. The data were obtained through a portable chlorophyll meter (MC-100, Apogee Instruments, Logan, UT, USA) equipment that determines a value proportional to the amount of chlorophyll present in the leaf [40].

Leaf area index (LAI; $m^2$ $m^{-2}$) was measured on a clear day at midday, after harvest, when plant growth had already stopped, for which an AccuPAR LP-80 ceptometer (Decagon Devices Inc., Washington, DC, USA) was used that delivers the average of 80 quantum sensors that determine direct, diffuse, residual and reflected photosynthetically active radiation from the soil [41].

### 2.5. Fruit Yield and Chemical Compounds

Fruit productivity (g plant$^{-1}$) was measured immediately after hand harvesting at 130 d after full flowering [6]. The period of full flowering took place at the beginning of September and the manual harvest was carried out at the end of December.

The total polyphenol content was determined using the Folin–Ciocalteu method following the indications by Romero-Román et al. [8], where first, an extract was obtained from the harvested sample, using 0.25 g sample, 2.5 mL $H_2O$:MeOH:formic acid solution (24:25:1) and taken to ultrasound CPX 5800 Branson (Branson Ultrasonics Corp., Danbury, CT, USA) for 1 h. Then, it was left for the rest of 24 h and taken for an ultrasound for 1 h, then; the sample was centrifuged for 15 min at 1209 g and filtered. Subsequently, a standard curve was prepared with gallic acid and then, the samples prepared with Folin–Ciocalteu reagent, distilled $H_2O$ and $Na_2CO_3$ 20% were mixed and incubated for 2 h in the dark, to be measured in the spectrophotometer at 760 nm [42]. The standard curve equation used was 0.0013x + 0.0573 and $R^2$ = 0.99. The results are expressed as mg gallic acid 100 $g^{-1}$ FW [43].

The DPPH antioxidant capacity was performed after Romero-Román et al. [44] by diluting the extract and incorporating DPPH solution, which was shaken and kept in the dark for 1 h, at room temperature, and then spectrophotometer readings were taken at 515 nm. Subsequently, a standard curve of Trolox was carried out [45]. The standard curve

equation used was $0.0008x + 0.0272$ and $R^2 = 0.99$. The results were expressed in µmol Trolox equivalent (TE) 100 $g^{-1}$ FW [6].

ORAC antioxidant capacity was performed by Romero-Román et al. [8]. Diluting 100 µL sample in phosphate buffer pH 7.4, fluorescein intensity was measured every 1 min for 1 h with excitation and emission wavelengths of 485 and 520 nm at 37 °C. The results were expressed as µmol TE 100 $g^{-1}$ FW [45].

Anthocyanins were identified by high-performance liquid chromatography with a diode array detector (HPLC-DAD) on a Hitachi primaide apparatus equipped with a photodiode array detector (Model 1430, Hitachi, Tokyo, Japan). The equipment consisted of a binary pump with a degasser (Model 1110, Hitachi, Tokyo, Japan) and an autosampler (Model 1430, Hitachi, Tokyo, Japan). The HPLC system was controlled by ChemStation software (version 08.03, Agilent Technologies, Palo Alto, CA, USA) [46]. Anthocyanin quantification was performed with the sample extracts used for the determination of the mentioned analyses, which were filtered through a 0.22 µm PVDF membrane (Millex V13, Millipore, Bedford, MA, USA). The results were expressed as mg 100 $g^{-1}$ FW [45].

## 2.6. Statistical Analysis

Data were subjected to analysis of variance (ANOVA) with a significance level of $p < 0.05$. Comparison of means was performed using Fisher's least significant difference (LSD) test with a significance level of 0.05, and in addition, principal component analysis (PCA) was performed using mean-centered data based on eigenvalues to determine the correlation and discrimination between soil, plant, and fruit variables under different water replenishment conditions using R software [47] with FactoMineR and ggplot2 packages [48].

## 3. Results

### 3.1. Biological Properties of Soil and Roots

Soil moisture monitoring showed that the treatment of 150% of $ET_0$ > 100% of $ET_0$ > 50% of $ET_0$ > 0% of $ET_0$, both in measurements taken 24 h after irrigation and 48 h after irrigation (Table 2).

Soil microbiological and enzyme activity showed significant differences among the different treatments (Table 3). The highest fluorescein diacetate (FDA) activity was reached with the treatment containing 100% $ET_0$, which presented 41.8 µg FDA $g^{-1}$, being significantly superior to the treatments containing 50% and 150% $ET_0$, which presented average values of 34.1 and 30.3 µg FDA $g^{-1}$, respectively, both significantly higher than the treatment without irrigation, which reached a value of 22.8 µg FDA $g^{-1}$ dry soil. It should be noted that irrigation treatments significantly influenced soil microbial respiration. The treatments with the highest irrigation dose, 100% and 150% $ET_0$, presented higher soil basal respiration with values of 2.2 and 2.1 µg $CO_2$ $g^{-1}$ $h^{-1}$, without significant differences between them, but significantly higher than 50% $ET_0$ treatment with 1.7 µg $CO_2$ $g^{-1}$ $h^{-1}$. It is important to note that in soil basal respiration, all irrigation treatments showed significant increase in comparison to the non-irrigated treatment.

**Table 3.** Soil microbiological properties and enzyme activity in response to water replenishment.

| Treatments | FDA Activity | Soil Basal Respiration | Urease Activity | Dehydrogenase Activity | Acid Phosphatase Activity | AMF Colonization in Roots |
|---|---|---|---|---|---|---|
| (ET$_0$ %) | (µg FDA g$^{-1}$) | (µg CO$_2$ g$^{-1}$ h$^{-1}$) | (µmol NH4$^+$ g$^{-1}$ h$^{-1}$) | (µg INTF g$^{-1}$) | (µmol PNP g$^{-1}$ h$^{-1}$) | (%) |
| 0 | 22.8 ± 1.70 c | 1.4 ± 0.10 c | 2.5 ± 0.09 b | 59.1 ± 0.72 c | 12.8 ± 0.48 b | 62.5 ± 4.79 a |
| 50 | 34.1 ± 1.27 b | 1.7 ± 0.08 b | 2.7 ± 0.06 ab | 109.8 ± 3.02 a | 14.6 ± 0.32 a | 82.5 ± 4.79 a |
| 100 | 41.8 ± 1.49 a | 2.2 ± 0.16 a | 2.5 ± 0.04 b | 83.9 ± 4.74 b | 14.0 ± 0.64 a | 65.0 ± 8.66 a |
| 150 | 30.3 ± 1.19 b | 2.1 ± 0.08 a | 2.8 ± 0.07 a | 76.0 ± 5.34 b | 7.7 ± 0.86 c | 72.5 ± 4.79 a |
| Anova *p*-Values | 0.0001 | 0.0007 | 0.047 | 0.0001 | 0.0001 | 0.1413 |

Treatments: 0%, 50%, 100%, and 150% ET$_0$. Different lowercase letters indicate significant differences between treatments according to Fischer's LSD test (*p* < 0.05). Mean ± standard error (*n* = 4). AMF: Arbuscular mycorrhizal fungi. Data on arbuscular mycorrhizal fungi (AMF) colonization on roots was analyzed by Kruskal–Wallis nonparametric analysis.

On the other hand, urease enzymatic activity increased significantly with the replacement treatment of 150% ET$_0$, presenting an increase of 12% with respect to 0% ET$_0$ treatment, which presented a value of 2.5 µmol NH$_4^+$ g$^{-1}$ h$^{-1}$.

As for dehydrogenase activity, the highest value reached was obtained with 50% replacement of ET$_0$, being 86% higher than the treatment without irrigation. Meanwhile, 100% and 150% ET$_0$ replenishment treatments with values of 83.9 and 76.0 µg INTF g$^{-1}$ presented increases of 42% and 29%, respectively, with respect to the treatment without irrigation.

Regarding acid phosphatase activity, irrigation of 50% and 100% ET$_0$, had increases of 14% and 9%, respectively, with respect to 0% ET$_0$. It should be noted that in acid phosphatase activity, the maximum irrigation rate applied to the crop (150% ET$_0$) was significantly lower than the rest of the treatments including the treatment without irrigation, registering a value of 7.7 µmol PNP g$^{-1}$ h$^{-1}$.

On the other hand, AMF colonization in roots (Figure 3a,b) did not show significant differences between treatments (*p* > 0.05).

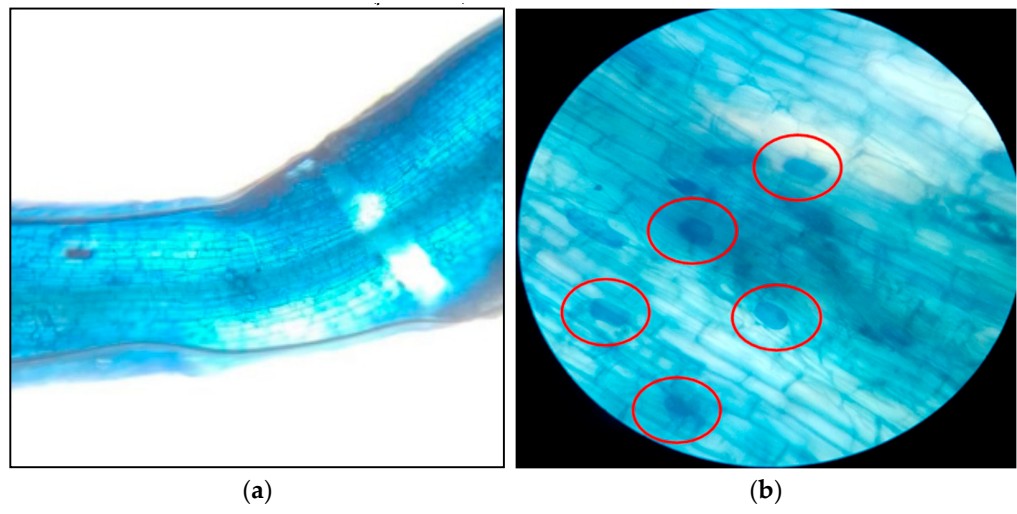

(a)                                                                              (b)

**Figure 3.** (**a**) Stained calafate fine root section, viewed microscopically at 10×; hydric replenishment rate of 50% ET$_0$. (**b**) AMF vesicles in stained calafate fine roots (structures circled in red), viewed microscopically at 40×; hydric replenishment rate of 50% ET$_0$. AMF: Arbuscular mycorrhizal fungi.

### 3.2. Plant Physiological Parameters

The maximum quantum yield of photosystem II (F$_v$/F$_m$; Figure 4a), measured at 09:00 and 12:00 h, did not show significant differences among treatments, reaching values close to 0.80 and 0.77, respectively. At 15:00 h, the 100%, 150% and 50% ET$_0$ replenishment rates had a 7%, 6% and 3% lower F$_v$/F$_m$ decrease than the 0% ET$_0$ treatment, which only reached 0.71 (Figure 4a). At 18:00 h, all treatments irrigated at 50%, 100% and 150% ET$_0$, reached a

mean $F_v/F_m$ value of 0.76, significantly higher than 0% $ET_0$ treatment, which only reached $F_v/F_m$ 0.72 (Figure 4a). Stomatal conductance (Figure 4b) presented significant statistical differences among treatments at 09:00 h, with 50% and 100% $ET_0$ irrigation presenting the greatest increases, 34% and 13%, respectively, with respect to the treatment without irrigation, which had a value of 128 mmol m$^{-2}$ s$^{-1}$. On the other hand, the irrigation of 150% $ET_0$ presented the lowest value of stomatal conductance measured at 09:00 h with 114 mmol m$^{-2}$ s$^{-1}$ ($p < 0.05$). In the subsequent measurements, 12:00, 15:00 and 18:00 h, nonsignificant differences were found between treatments, with mean values of stomatal conductance close to 133, 91 and 101 mmol m$^{-2}$ s$^{-1}$, respectively for each of the aforementioned times (Figure 4b).

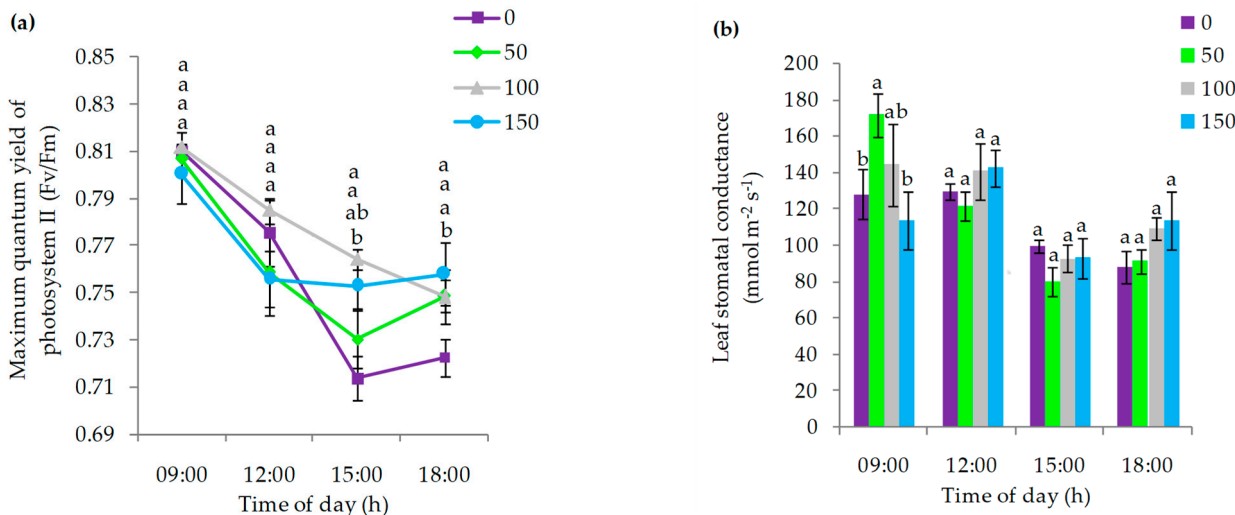

**Figure 4.** (**a**) Variation of the maximum quantum yield of photosystem II ($F_v/F_m$) in calafate plants; (**b**) Values recorded for stomatal conductance (mmol m$^{-2}$ s$^{-1}$), in calafate plants subjected to different doses of water replenishment, evaluated at different times of the day: 09:00, 12:00, 15:00 and 18:00 h. Treatments: 0%, 50%, 100% and 150% $ET_0$. Different lowercase letters indicate significant differences between treatments according to Fischer's LSD test ($p < 0.05$). Mean ± standard error ($n = 4$). Bars correspond to experimental error for each treatment.

The Leaf area index (LAI) was not influenced by the water replenishment treatments ($p > 0.05$), all of them reaching a mean value of LAI close to 2.2 m$^2$ m$^{-2}$ (Figure 5a). Leaf chlorophyll index was influenced by irrigation dose ($p < 0.05$), where 50% $ET_0$ replenishment significantly increased SPAD (Soil Plant Analysis Development) value by 45% with respect to the 0% $ET_0$ treatment. On the other hand, the treatments with higher irrigation, replenishment of 100% and 150% $ET_0$, without statistical differences, increased their SPAD value by 33% with respect to the treatment without irrigation (Figure 5b).

### 3.3. Fruit Yield and Chemical Parameters

Fruit yield per plant was significantly higher with 50% $ET_0$ irrigation ($p < 0.05$) compared to the rest of the treatments, reaching a mean value of 359 g plant$^{-1}$, while the treatments of 0%, 150% and 100% $ET_0$ replenishment registered mean values of 253, 226 and 193 g plant$^{-1}$, respectively, with nonsignificant differences among them (Figure 6a). It should be noted that irrigation treatments had a significant influence on the total polyphenol content of the fruit ($p < 0.05$). The treatments without irrigation and with an irrigation of 150% $ET_0$, reached the highest values recorded, being these 920 and 1001 mg gallic acid 100 g$^{-1}$ fresh weight (FW), respectively. On the other hand, the irrigation treatment of 100% $ET_0$ with 830 mg gallic acid 100 g$^{-1}$ FW was superior to the treatment with 50% $ET_0$ ($p < 0.05$), which presented a value of 620 gallic acid 100 g$^{-1}$ FW (Figure 6b).

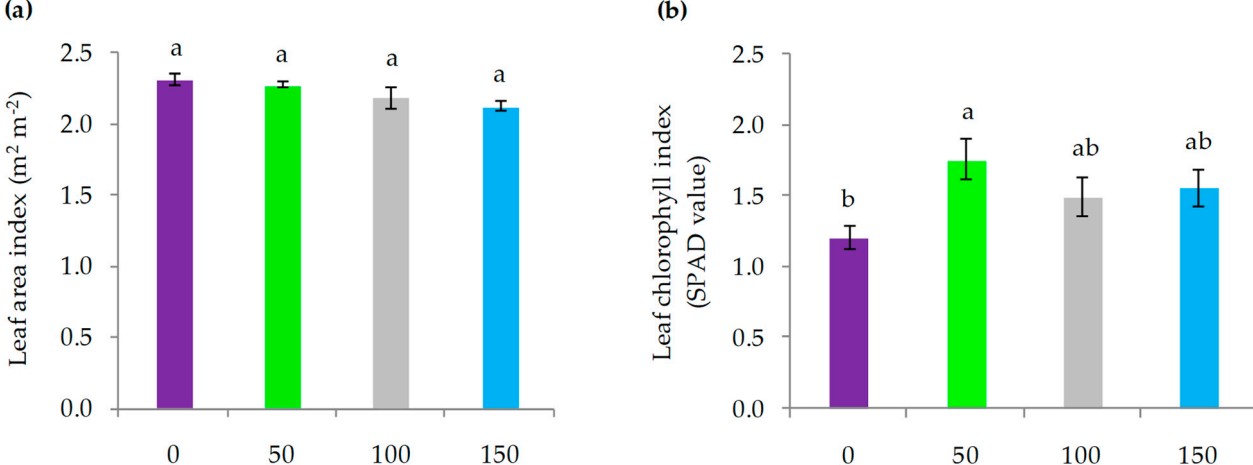

**Figure 5.** (**a**) Leaf area index values recorded in calafate plants; (**b**) leaf chlorophyll index recorded in calafate leaves subjected to different doses of water replenishment. Treatments: 0%, 50%, 100% and 150% $ET_0$. Different lowercase letters indicate significant differences between treatments according to Fischer's LSD test ($p < 0.05$). Mean ± standard error ($n = 4$). Bars correspond to experimental error for each treatment.

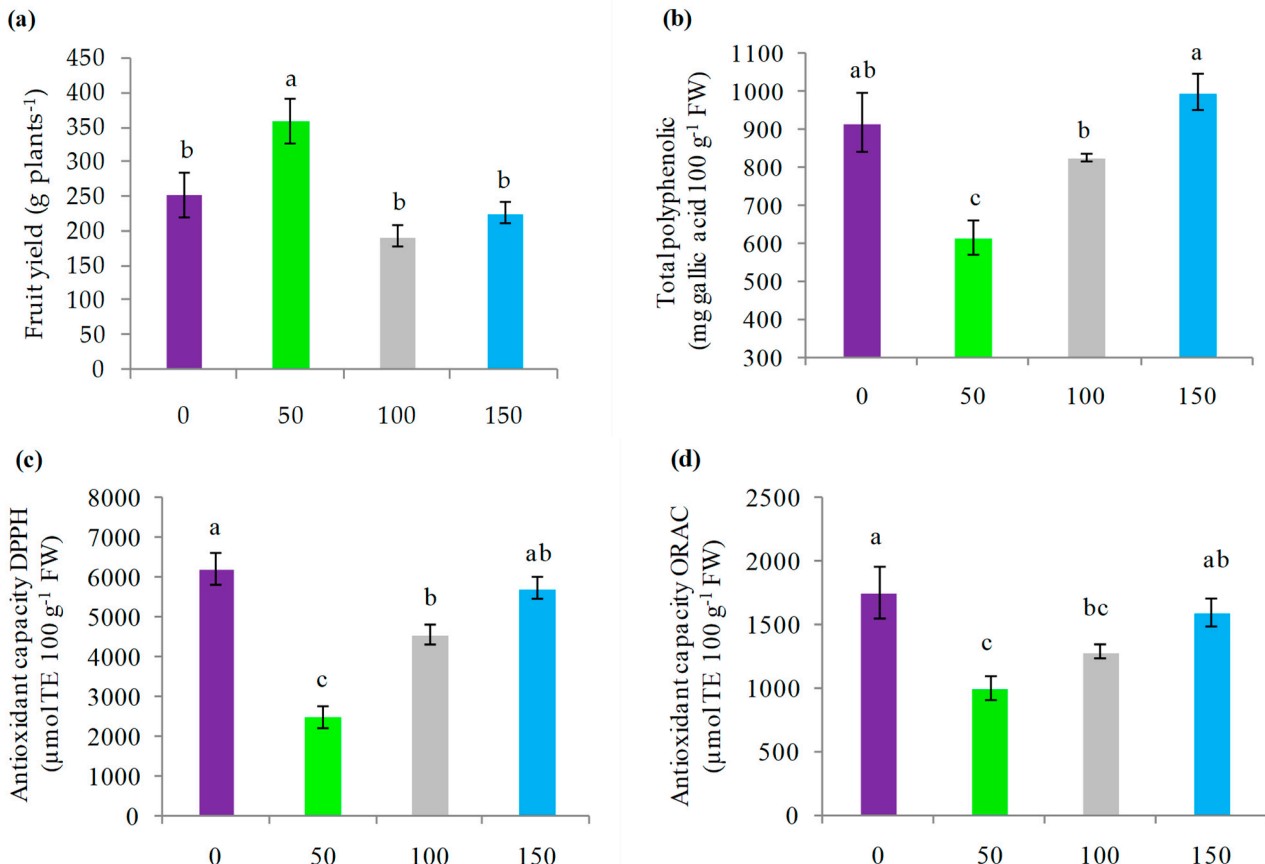

**Figure 6.** (**a**) Average yield of fresh fruits of calafate (g plant$^{-1}$); (**b**) total polyphenols; (**c**) DPPH antioxidant capacity; and (**d**) ORAC antioxidant capacity in fruits of calafate plants subjected to different doses of water replenishment. Treatments: 0%, 50%, 100% and 150% $ET_0$. Different lowercase letters indicate significant differences between treatments according to Fischer's LSD test ($p < 0.05$). Mean ± standard error ($n = 4$). Bars correspond to experimental error for each treatment.

Likewise, treatments with irrigation significantly influenced the antioxidant capacity 2,2-diphenyl-picryl-hidrazyl (DPPH) (Figure 6c) and oxygen radical absorbance capacity (ORAC) (Figure 6d). The treatments without irrigation and with 150% $ET_0$ replenishment presented the highest DPPH values corresponding to 6236 and 5764 µmol TE 100 g$^{-1}$ FW, which were followed by the 100% $ET_0$ hydric replenishment treatment, which reached an average value of 4592 µmol TE 100 g$^{-1}$ FW, being significantly higher than 50% $ET_0$, which presented 2528 µmol TE 100 g$^{-1}$ FW. Likewise, the ORAC antioxidant capacity was higher with the 0% and 150% $ET_0$ replenishment treatments, with values of 1757 and 1609 µmol TE 100 g$^{-1}$ FW, respectively (Figure 6d). On the other hand, 50% and 100% $ET_0$ replenishment did not present significant differences between them ($p > 0.05$), with ORAC antioxidant capacity values of 1011 and 1297 µmol TE 100 g$^{-1}$ FW, respectively. As for the total anthocyanin content, it presented the same trend as the ORAC antioxidant capacity, with the 0% and 150% $ET_0$ replenishment treatments presenting the highest mean content with 610.75 and 484.68 mg 100 g$^{-1}$ FW, respectively, followed by 50% and 100% $ET_0$ with 286.37 and 468.26 mg 100 g$^{-1}$ FW, respectively, which did not present significant differences between them. The total anthocyanin content was represented by 82% to 85% by three main anthocyanins: delphinidin, petunidin and malvidin 3-glucoside (Table 4).

**Table 4.** Anthocyanins (mg 100 g$^{-1}$) of fresh calafate fruit by HPLC according to water replenishment treatments.

| Anthocyanins | Treatments ($ET_0$%) | | | | Anova p-Values |
|---|---|---|---|---|---|
| | 0 | 50 | 100 | 150 | |
| Petunidin 3,5-dihexoside | 13.7 ± 2.4 a | 6.6 ± 0.8 b | 12.1 ± 0.8 ab | 16.5 ± 1.7 a | 0.0018 |
| Malvidin 3,5-dihexoside | 13.6 ± 1.9 a | 5.4 ± 0.9 b | 6.8 ± 0.7 b | 12.0 ± 1.6 a | 0.0013 |
| Delphinidin 3-glucoside | 208.7 ± 27.2 a | 112.6 ± 21.3 b | 201.2 ± 15.1 ab | 196.6 ± 30.5 ab | 0.0277 |
| Delphinidin 3-rutinoside | 6.1 ± 1.0 a | 3.9 ± 0.8 a | 3.8 ± 1.0 a | 4.2 ± 1.0 a | 0.1741 |
| Cyanidin 3-glucoside | 37.9 ± 6.7 a | 12.0 ± 2.1 b | 23.8 ± 3.3 ab | 36.1 ± 6.8 a | 0.004 |
| Petunidin 3-glucoside | 154.3 ± 16.9 a | 77.3 ± 14.7 b | 126.3 ± 8.7 ab | 125.9 ± 13.3 ab | 0.0048 |
| Petunidin 3-rutinoside | 8.5 ± 1.6 a | 4.5 ± 1.0 b | 5.1 ± 0.7 b | 5.2 ± 0.9 b | 0.0297 |
| Peonidin 3-glucoside | 29.5 ± 4.15 a | 6.1 ± 0.96 b | 12.1 ± 1.62 b | 11.8 ± 2.00 b | 0.0001 |
| Malvidin 3-glucoside | 138.4 ± 10.1 a | 54.3 ± 6.7 b | 66.5 ± 8.6 b | 79.7 ± 8.5 b | 0.0001 |
| Total anthocyanins | 610.8 ± 61.1 a | 286.4 ± 46.3 b | 468.3 ± 24.4 ab | 484.7 ± 55.3 a | 0.0016 |

Treatments: 0%, 50%, 100% and 150% $ET_0$. Different lowercase letters indicate significant differences between treatments according to Fischer's LSD test ($p < 0.05$). Mean ± standard error ($n = 4$).

*3.4. Correlations*

The correlation matrix (Figure 7) indicates that soil FDA was negatively correlated with fruit DPPH antioxidant activity; the higher the FDA, the lower the DPPH antioxidant activity. Likewise, dehydrogenase activity was negatively related to total polyphenol content, total anthocyanins, and fruit DPPH and ORAC antioxidant capacity. On the other hand, acid phosphatase activity was negatively correlated with fruit total polyphenol content. Among the soil parameters measured, FDA was positively correlated with soil respiration and urease activity was negatively correlated with acid phosphatase activity. Dehydrogenase activity was positively correlated with soil respiration and acid phosphatase activity was positively correlated with AMF colonization. On the other hand, the leaf area index (LAI) of the plant was positively correlated with total polyphenol content, total anthocyanins and antioxidant activity ORAC of fruits. The correlation between fruit parameters was positive for antioxidant capacity DPPH and ORAC, with total polyphenol content. In addition, fruit total anthocyanin content was positively correlated with total polyphenol content and antioxidant activity DPPH and ORAC. In contrast, fruit yield was negatively correlated with fruit total polyphenol and total anthocyanin content (Figure 7).

Principal component analysis (PCA) (Figure 8a) was performed for 13 parameters: arbuscular mycorrhizal fungi, soil microbiological activity, soil microbial respiration, urease activity, dehydrogenase activity, acid phosphatase activity, LAI, chlorophyll index, fruit

yield, total polyphenols, DPPH antioxidant capacity, ORAC antioxidant capacity and total anthocyanins. The principal components PC1 and PC2 retained 44.5% and 18.7%, respectively. This represents all parameters as vectors in the biplot, while the vector length shows how well-represented the variables are in this plot. The treatments in the PCA (Figure 8b) are represented by the numbers 1–4 for the 0% $ET_0$; 5–8 for the treatment with 50% $ET_0$; 9–12 for the treatment with 100% $ET_0$; and 13–16 for the treatment with 150% $ET_0$. These results confirm what was previously indicated in the correlation matrix.

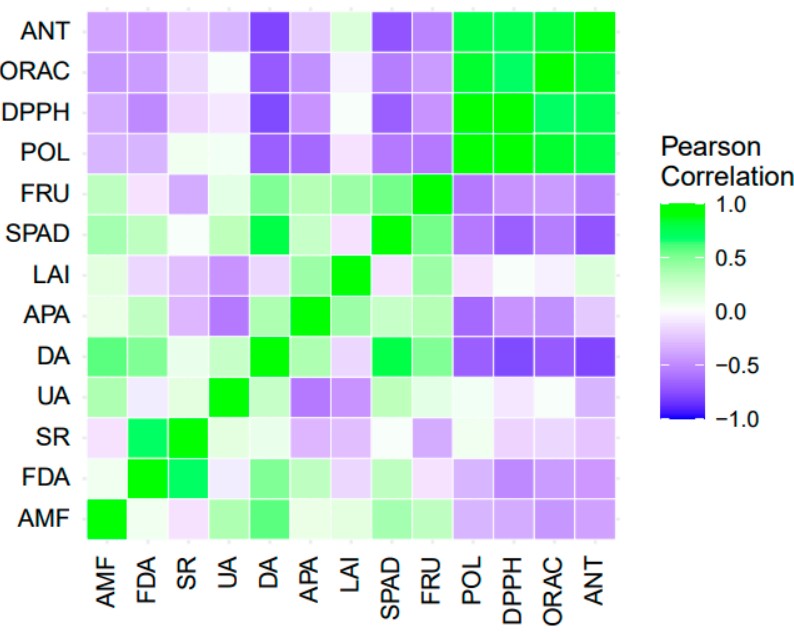

**Figure 7.** Correlation matrix for soil variables, plant physiological parameters, yield parameters and fruit bioactive compounds. AMF: Arbuscular mycorrhizal fungi; FDA: soil microbiological activity; SR: soil microbial respiration; UA: urease activity; DA: dehydrogenase activity; APA: acid phosphatase activity; LAI: leaf area index; SPAD: chlorophyll index; FRU: fruit yield; POL: total polyphenols; DPPH: DPPH antioxidant capacity; ORAC: ORAC antioxidant capacity; and ANT: total anthocyanins.

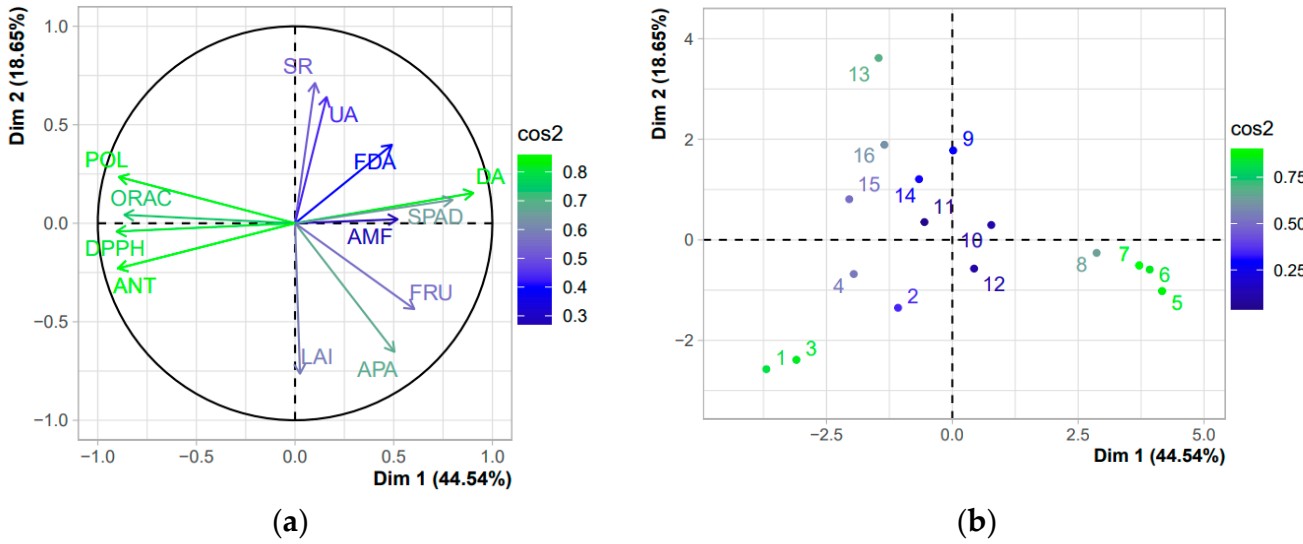

**Figure 8.** Principal component analysis (PCA) of arbuscular mycorrhizal fungi (AMF); soil microbiological activity (FDA); soil microbial respiration (SR); urease activity (UA); dehydrogenase activity (DA); acid phosphatase activity (APA); leaf area index (LAI); chlorophyll index (SPAD); fruit yield (FRU); total polyphenols (POL); DPPH antioxidant capacity (DPPH); ORAC antioxidant capacity (ORAC); and total anthocyanins (ANT). (**a**) PCA of variables; (**b**) PCA of individuals.

## 4. Discussion

### 4.1. Microbiological Parameters of Soil and Roots

Soil microbial communities and their activity define soil quality, as they carry out fundamental processes related to nutrient cycling and biodegradation of soil organic compounds [49]. In the present investigation, soil microbial activity, measured as FDA and soil microbial respiration, was influenced by irrigation dose, with significant increases ($p < 0.05$) in treatments irrigated with 50%, 100% and 150% $ET_0$. In agreement with the present results, previous studies have shown that water deficit through frequent or intermittent drought reduces soil microbial activity and modifies community composition, as microbes focus on synthesizing osmolytes and intracellular maintenance strategies [50]. On the other hand, irrigation increases the availability of organic matter due to release by physical disruption of soil aggregates and/or cell lysis, thus increasing microbial growth and activity [51]. Therefore, we can infer that the irrigation doses that had the greatest effect on soil microbial activity (50% and 100% $ET_0$) increased soil water potential, which produced a decrease in the compensatory intracellular solute concentrations of soil microorganisms, accelerating biochemical functions and preventing cytoplasmic desiccation [52]. This is also consistent with what was demonstrated in a grapefruit orchard, where irrigation dose produced changes in the relative abundance of soil microbial populations at the phylum level, where irrigation at 100% $ET_0$ increased the relative abundance of Firmicutes, while an irrigation corresponding to 50% $ET_0$ increased the abundance of Proteobacteria [24]. The lower increases in FDA by irrigation at 150% $ET_0$ indicate that this irrigation dose probably reduces soil aeration [53], since it was shown that soil moisture (%) of the treatment with 150% of $ET_0$ (Table 2) in the months of maximum demand was up to 35% after 24 h past one irrigation, which, corresponded to half the soil porosity of the study site, which was 0.67% (data not shown). However, the high percentage of organic matter present in the study site soil (9.7%), led to better structural conditions for air circulation and water drainage, reaching soil microbial respiration values up to 50% higher than the non-irrigated treatment [53]. Although our finding has not been previously described in cultivated and wild calafate, we believe it is necessary to evaluate the effects of irrigation dose by analyzing the relative abundance of soil microorganism communities in future research.

Soil microorganisms synthesize and secrete extracellular enzymes, which play an important role in soil nutrient cycling [54]. In our study, the irrigation rates evaluated affected differentially the activity of the quantified enzymes. The observed increase in urease activity with water replenishment of 50% $ET_0$ could be attributed to a higher soil microbial mass produced under this treatment capable of hydrolyzing soil organic matter (SOM) (9.7%) containing N [55], with the site soil having ideal conditions by presenting high SOM content with 9.7% and low available N content (19 mg kg$^{-1}$). These results agree with those reported in a Mediterranean forest of *Quercus ilex* L., where increasing the intermediate soil moisture by 10% and 21%, also increased the activity of this enzyme up to 10% and 42%, respectively [56]. We can also infer that the irrigation dose with 150% $ET_0$ probably led to reduced high soil temperatures during the time of irrigation application, since it has been shown that urease activity not only depends on soil fertility and water availability, but climatic factors such as soil temperature determine the phenology of enzymes such as urease [57]. Dehydrogenase is an extracellular enzyme that plays an important role in the oxidation of organic matter and the incorporation of soil organic C [58]. In our study, dehydrogenase activity was 86% higher with respect to the non-irrigated treatment under 50% $ET_0$ irrigation. In agreement with the present results, previous studies have shown that soil drought significantly decreases the action of this enzyme [59], resulting in a slowing down of SOM decomposition [60]. In contrast, adequate soil moisture through irrigation significantly increases its activity [60]. Irrigations with higher doses (100 and 150% $ET_0$), followed a pattern similar to that of FDA and soil microbial respiration, which is explained by the fact that this enzyme only exists in viable microbial cells and is a good indicator of the global metabolic activity of the soil [59]. Acid phosphatase can hydrolyze organic P to inorganic P compounds [61]. The results obtained show that acid phosphatase

activity was slightly higher (14% and 9%) with irrigations of 50% and 100% $ET_0$ with respect to the treatment without irrigation. It has been shown that a drought condition decreases the activity of this enzyme due to factors such as, lower secretion by soil microbes, lower stabilization efficiency by the smaller SOM, and even, high temperature conditions can denature this enzyme [59]. Strangely, the 150% $ET_0$ decreased and inhibited the activity of this enzyme. Recently, it has been shown that high-dose irrigation was able to decrease the activity of this enzyme in an almond orchard [25]. However, further work is needed to clarify the dynamics of this enzyme by irrigation dose effect in fruit orchards, especially in recently established species such as calafate.

Associations between roots of dicotyledonous plants and AMF are more common than in the roots of non-mycorrhizal plants. This association has already been reported in some species of the family Berberidaceae, for example, in the species *Berberis vulgaris* L. [62]. This association has also been reported in *Berberis* species from southern Chile, being the percentage of colonization similar to that reported in our study [63]. Although there were nonsignificant differences between irrigation treatments, a higher colonization percentage (20%) was determined for the irrigation treatment with 50% $ET_0$ compared to the non-irrigation treatment. It has been shown that AMF colonization in Mediterranean climates is influenced by soil nutrient fertility such as P [64]. Therefore, we believe that the 50% $ET_0$ irrigation dose, would increase somewhat the colonization capacity of these microorganisms by presenting a soil with better structural conditions [53] and P fertility mediated by acid phosphatase [61], leading to competitive exclusion of other soil microorganisms [64]. However, we recognize that our results may be somewhat limited by measuring only a single parameter corresponding to AMF colonization. Further work is needed to taxonomically characterize AMF using molecular techniques to identify their diversity and population structure in order to better understand the ecological impacts of irrigation dose on the calafate crop.

### 4.2. Plant Physiological Parameters

Irrigation dose influenced the indirect physiological parameters measured in this study. The maximum quantum yield of photosystem II had a decreasing trend as the day progressed. This is consistent with that reported in a calafate orchard in south-central Chile under the same radiation conditions, which reached maximum direct photosynthetically active radiation values close to 2000 $\mu$mol m$^{-2}$ s$^{-1}$. Research has also shown that the irrigation treatment with 50% $ET_0$ recovers earlier and with higher values of $F_v/F_m$ its photosynthetic apparatus with respect to the treatment without irrigation. This increase could be attributed to better physiological and nutritional conditions of the plant to cope with abiotic stress factors such as high levels of radiation and temperature that can generate photooxidative damage to the leaves [65]. These results also corroborate the findings of Retamal-Salgado et al. [38], who demonstrated that high ambient radiation and temperature can negatively affect the efficiency of the photosynthetic apparatus in blueberries. Another important finding in the present investigation is that 50% and 100% $ET_0$ treatments positively influenced leaf stomatal conductance, presenting higher values at the beginning of the day. Plant irrigation has been shown to drastically influence growth, development, and physiological characteristics of photosynthesis [66]. Unlike these treatments, the 150% ET0 treatment did not respond in the same way, which may be due to oxygen depletion in the root zone affecting physiological functions of the plants, since, as shown in Table 2, this treatment after 24 and 48 h of irrigation showed higher soil moisture than the other treatments. In blueberries of the cultivars highbush and rabbiteye, it was shown that irrigation doses above 100% of ET0 are capable of producing oxygen depletion in the root zone, leading to a decrease in stomatal conductance and photosynthesis, in addition, they determined, for both processes to recover to optimal conditions reached prior to irrigation, long periods of up to 18 days or more were required [67] which does not occur in our study, being the period between irrigations of two days. Therefore, we believe that in responses to stress conditions (Treatments 0% and 150% $ET_0$) gas exchange may decrease and even close

their stomata to avoid water loss [68]. This is consistent with that reported in a rabbit's eye blueberry orchard [68].

Irrigation rate affected leaf chlorophyll content ($p < 0.05$). Irrigation treatments increased chlorophyll levels in leaves compared to the non-irrigation treatment. Irrigation of 50% $ET_0$, had greater increases in leaf chlorophyll. Soil moisture has been shown to alter plant physiological response by improving soil physicochemical conditions for plant nutrient transport and utilization and increasing chlorophyll content to promote photosynthesis and respiration [68]. We can also infer that stressed plants with lower chlorophyll values have less photosynthesis and respiration, which is corroborated by the previously discussed data of photosystem II maximum quantum yield and stomatal conductance. The leaf area index is an indirect indicator of the physiological state of the plant; which in our case did not respond to the irrigation dose. These results corroborate, but extend, the findings of much of the previous work on wild calafate [65,69,70] that demonstrate the hardiness of the plant and its high degree of phenotypic adaptation. Consequently, this is the first time that this approach has been proposed to determine the influence of irrigation dose on the canopy of domesticated calafate. This can be considered a significant advance in its level of adaptation to drought conditions and excessive soil moisture.

### 4.3. Fruit Yield and Chemical Parameters

The effect of irrigation dose has been studied in different fruit species because of its importance in plant productivity [14]. Irrigation at 50% $ET_0$ produced the highest fruit yield compared to the other treatments ($p < 0.05$). This is consistent with the literature, which reported similar yields under the same temperature conditions, mean 14.5 °C and maximum 31 °C, and Andisol origin, as well as the same harvest date defined at 130 days after full flowering [6]. In blueberries, when evaluating different irrigation doses, it was shown that an intermediate irrigation dose of 100% $ET_0$ produced significant increases in different productive parameters such as fruit weight, number of fruits per plant and fruit yield, because the plants presented less stress [67], a situation similar to that of the 50% $ET_0$ treatment in our research. In our study, the yield of plants irrigated with a dose of 0%, 100% and 150% $ET_0$ was lower than the treatment of 50% $ET_0$ with nonsignificant differences between them (Figure 6a). Consistent with the present results, previous studies have shown that there are significant reductions in fruit yield in some berries due to severe drought conditions [71] and excessive irrigation [72] that cause stress on the plant by inducing stomatal closure, decreased $CO_2$ assimilation, decreased root activity and increased prevalence to fungal diseases [70]. However, our results show that calafate plants under different degrees of water stress with doses of 0% and 150% $ET_0$, are able to fruit similarly, which corroborates its high hardiness described for different climatic conditions with contrasting rainfall levels [7]. According to our results (Figure 6a), it was shown that the irrigation treatment with 0% of ET0 did not present significant statistical differences with the treatments 100% and 150% of ET0, which demonstrates that the water requirements of calafate to generate its physiological functions and fruiting are low. Our results are corroborated by Radice et al. [7] who demonstrated that non-irrigated cauliflower in the region of Moreno, Argentina, with warm temperate climates similar to those of our study were able to fruit, however, the lower water regimes of Moreno compared to Ushuaia, Argentina, produced smaller fruit size. Therefore, we emphasize the importance of evaluating physical parameters of the fruit in future research. Furthermore, to our knowledge, this is the first study published to date that demonstrates the effects of irrigation dose on the productivity of cultivated calafate. However, we recognize that our results may be somewhat limited by seasonality, because only one season was evaluated and a single productivity parameter was determined.

Phenolic compounds confer the particular characteristics of color and aroma in calafate and other berries such as blueberries, raspberries, and strawberries [67], with anthocyanins being responsible for defining color during the veraison stage until harvest [8]. Higher concentrations of polyphenols and anthocyanins were observed in the most stressed treat-

ments of the present study, both due to water deficiency and excess water replenishment. Previous studies have shown a high accumulation of anthocyanins influenced by abiotic factors such as water stress [73], which is part of the protective mechanism of plants to cope with oxidative stress through the uptake of reactive oxygen species [74]. For their part, Bryla et al. [72] demonstrated similar stress effects in blueberry plants by high doses of water replenishment. The decrease in the concentrations of polyphenolic compounds observed in the 50% $ET_0$ treatment could be attributed to the higher fruit yield, which may result in larger caliber and delayed physiological maturity of the fruit [71], which could have diluted the content of bioactive compounds in the fruit due to increased water content. In this sense, a study on *Vitis vinifera* L. showed that the irrigation dose increased fruit yield but decreased the content of polyphenolic compounds [75].

Research has also shown that the predominant anthocyanins were delphinidin, petunidin and malvidin and their derivatives, similar to that reported by Romero-Román et al. [8] in wild calafate from southern Chile. This is also consistent with that reported in grapes, where the three glycoside-conjugated anthocyanins are the majority due to the temperature and pH of the berry extract [76]. Delphinidin represented between 35% and 41% of the total anthocyanins, being lower with the treatment without water replenishment. This decrease could be attributed to the low stability of anthocyanins to external factors such as light and temperature, which were probably more intense when there was no irrigation, a situation similar to that reported by Torres et al. [77], who demonstrated in grapes that water stress decreased pH of the fruit and, as a result, the fruit turned reddish instead of purple. The results of total delphinidins and anthocyanins reported in this study are lower than those previously reported in wild calafate [8]. This could be explained by differences in the quality of solar radiation in both agroclimatic regions [78], since in the study region it reaches a value of photosynthetically active radiation 26% higher in southern Chile [6]. Samkumar et al. [79] found that red light produced a higher induction of specific anthocyanin and delphinidin biosynthesis genes. We can also infer that the time of harvest is a determining factor in the accumulation of anthocyanins by fruit ripening [80] being in our case at 130 days after full flowering during December. Our delphinidin results are lower than those (80%) reported in maqui (*Aristotelia chilensis* (Mol.) Stuntz), which is another Chilean native berry species with high anthocyanin content in fruit [80] but which are similar to the delphinidin accumulation of 40% reported in blueberries [81]. Another important finding in the present investigation is that total antioxidant activity was influenced by irrigation dose, being higher in treatments of 0% and 150% $ET_0$. These values are similar to those reported in previous research under wild conditions [8,82]. However, irrigation doses with 50% and 100% $ET_0$ showed lower values of fruit antioxidant capacity but similar to those reported in cultivated calafate from central-southern Chile [6]. The anthocyanin content may explain the total antioxidant activity observed in the fruit ($r = 0.89$), a situation similar to that previously reported in maqui from central-southern Chile [80]. We can highlight that the irrigation treatment of 50% $ET_0$, although it presented a fruit yield 86% higher than the irrigation treatment of 100% $ET_0$, it maintained its quality, by presenting a significantly similar value of total anthocyanins and ORAC antioxidant capacity. To our knowledge, this is the first report on the effects of irrigation on anthocyanins and their antioxidant capacity in cultivated calafate fruit.

### *4.4. Influence of Variables*

The correlation matrix reaffirmed the importance of evaluating water replenishment management in a soil-plant system and chemical compound content of calafate fruit. Although moderate correlations were found between soil microbiological parameters and two plant physiological parameters measured in the correlation matrix, a more significant positive correlation ($r = 0.78$) was found between dehydrogenase and plant chlorophyll index (Figure 7). It has been previously reported that dehydrogenase plays an important role in the incorporation of soil organic C under soil water replenishment treatments that directly affects the physiological state of the plant through its nutrition [58]. Likewise,

dehydrogenase was one of the most sensitive enzymes to irrigation on fruit chemical composition, demonstrated by its high negative correlation with parameters such as total anthocyanins ($r = -0.79$), total polyphenols ($r = -0.69$), antioxidant capacity DPPH ($r = -0.78$) and ORAC ($r = -0.71$). On the other hand, FDA and AMF colonization were directly and negatively correlated with the evaluated fruit chemical parameters (Figure 7). Recently, a study in almond trees showed that the higher soil microbial activity produced by an optimal irrigation dose negatively affects fruit chemical compounds [24], which could be explained by the higher yield and fruit size resulting from higher values of measured soil microbial activity and plant physiological parameters. Another important finding is the fact that irrigation directly influences the content of bioactive compounds in the fruit. The high relationship between the bioactive compounds of the fruit corroborate the above, for example, there was a high positive correlation between total polyphenols and antioxidant capacity measured through different modes of action, DPPH ($r = 0.89$) and ORAC ($r = 0.83$), which agrees with what has been previously reported in maqui from central-southern Chile [80]. The principal component analysis of the variables (Figure 8a) corroborates the above, since there is a greater closeness in distance and colors between the soil, plant and fruit variables mentioned above.

Through the principal component analysis of the variables (Figure 8a,b), the research shows that the most stressed treatments (0% and 150% $ET_0$) are the ones that presented a positive correlation with the bioactive compounds of the fruit, but not the treatments of 50% and 100% $ET_0$, which presented a positive correlation with the microbiological and enzymatic activities of the soil and physiological and productive activities of the plant. These results can be considered a significant advance in the domestication of calafate through the irrigation dose, capable of influencing the soil microbiological communities, which are closely related to the plant physiology and its productivity of fruits and/or bioactive compounds. Furthermore, to our knowledge, this is the first study published to date that demonstrates the influence of irrigation on soil microbiology, the calafate plant, and its productivity.

### 5. Conclusions

Our study was able to demonstrate that irrigation of 50% $ET_0$ significantly increased urease, dehydrogenase and soil acid phosphatase, which led to greater physiological response of the plant with higher values of stomatal conductance and chlorophyll index. This also led to a significant increase in fruit production, which maintained an adequate level of total anthocyanins and ORAC antioxidant capacity, being similar to the 100% $ET_0$ irrigation. On the contrary, 0% and 150% $ET_0$ treatments showed the lowest values in the mentioned parameters, but higher values on total anthocyanins and antioxidant capacity of the fruit. Based on these results, we can conclude that although the calafate plant has a high adaptability to extreme conditions of water replenishment such as drought and over-watering, it shows better results at soil-plant and fruit level with an irrigation of 50% $ET_0$, allowing the optimization of water resources and thus contributing efficiently to food security. On the other hand, as calafate is a very rustic species, we propose to evaluate in future research suboptimal irrigation doses between 0% and 50% $ET_0$, to further improve water efficiency and evaluate the chemical composition of the fruit. We emphasize soil microbiological evaluations should not be neglected, to better understand their contribution to the adaptability of this species under commercial orchards.

**Author Contributions:** Conceptualization, J.R.-S. and M.S.; methodology, J.R.-S., M.S., M.B. and R.V.-R.; software, M.B. and M.D.L.; validation, J.R.-S., M.S. and M.D.L.; formal analysis, J.R.-S. and M.B.; investigation, M.B., M.S. and J.R.-S.; resources, M.S., M.D.L. and J.R.-S.; data curation, M.D.L., J.R.-S. and M.B.; writing—original draft preparation, M.B., M.S., J.R.-S., R.V.-R. and M.D.L.; writing—review and editing, J.R.-S., M.S., M.D.L. and R.V.-R.; project administration, M.S. and J.R.-S.; funding acquisition, M.S., J.R.-S., R.V.-R. and M.D.L. All authors have read and agreed to the published version of the manuscript.

**Funding:** This research was funded by research project Nº 98 and 106 were granted by the Universidad Adventista de Chile, Chile, and Comisión Nacional de Investigación Científica y Tecnológica (CONICYT, Chile) scholarship (21201481/2020).

**Institutional Review Board Statement:** Not applicable.

**Informed Consent Statement:** Not applicable.

**Data Availability Statement:** The data presented in this study are available in the article.

**Acknowledgments:** Doctoral Program in Agronomy Sciences, from the University of Concepción. Recognition and gratitude to the technicians of the Chemical Analysis Laboratory, Department of Plant Production-University of Concepción, Chillán, thesis students and undergraduate students of the Adventist University of Chile for their contribution to the physiological measurements of plants and fruit harvest.

**Conflicts of Interest:** The authors declare no conflict of interest.

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
