# Peer review of "Plant Performance and Soil Microbial Responses to Irrigation Management: A Novel Study in a Calafate Orchard"

_horticulturae, doi:10.3390/horticulturae8121138_

Round 1

Reviewer 1 Report

The problem that the authors decided to solve is an important environmental one, especially in the light of climate changes which can affect food availability as well as its quality. Data on response of Berberis to different soil water availability is scarce in the literature.

The introduction clearly presents the problem and goals the authors want to achieve with the paper. The goal stated by the authors is of high significance.

Material and methods

The description of methods applied in the study is correct. Some supplementations and explanations should be added:

125- 128. Description of ETo calculation is insufficient (despite the reference cited). It is crucial for full understanding the irrigation regimes applied. More detailed information is required.

In future work, it would be beneficial to present climatic water balance (ET vs precipitation) to visualize agroclimatical situation better during the study period.

134-136. Soil moisture measurements using Diviner 2000 system. No data is presented and it should be mentioned here.

Results and Discussion

The description of results, also in terms of graphic design, may be accepted. The discussion explains the meaning of the results exhaustively. It contains reference to many items of the worldwide literature. On the basis of the results analysis, the authors drew clear conclusions and even practical preliminary suggestions.

Author Response

Dear Reviewer 1: Thank you very much for your suggestions and comments, they will help improve the quality of the manuscript.

Remark 1: 125- 128. Description of ETo calculation is insufficient (despite the reference cited). It is crucial for full understanding the irrigation regimes applied. More detailed information is required.

Response 1: The request was incorporated, including details of the applied irrigation systems.

Remark 2: In future work, it would be beneficial to present climatic water balance (ET vs precipitation) to visualize agroclimatical situation better during the study period.

Response 2: Future works will include the climatic water balance. However, Figure N°2 shows the water balance during the irrigation period.

Remark 3: 134-136. Soil moisture measurements using Diviner 2000 system. No data is presented and it should be mentioned here.

Response 3: The request was incorporated in Table 2.

Reviewer 2 Report

The article presents the results obtained by applying four irrigation treatments on soil microbiological activity, plant physiological response, fruit yield and chemical composition in a calafate orchard. The results are interesting; It should be noted that studies of this type are not very common. It is however necessary that the authors describe in a better way how the reference evapotranspiration ET0 was calculated, because its value was used as a basis for the definition of the irrigation treatments. It is also necessary that the authors present in detail the temporal evolution of the soil moisture content in each treatment. This last information is important since, as presented by the authors in the manuscript, almost all the microbiological and physiological variables of the crop depend on the soil moisture content. Additionally, this information can be used by the authors as the basis for many of their conclusions.

More remarques below.

2. Materials and Methods

2.1 Orchard establishment and edaphoclimatic characteristics of the study site

Line 101: “…and accumulated rainfall of 521.1 mm concentrated…”; …and annual accumulated rainfall of 521.1 mm…

Line 108: “The total number of plants in the orchard was 352…”; on what surface?

2.2 Experimental setup of the test

Lines 125 to 128: “To determine the actual daily irrigation, ET0 was calculated from the mean values of the previous 5 years taking as a reference what was suggested by Romero et al. [28], with data obtained from the agroclimatic station of INIA Quilamapu located near the study site [29]”; Considering that the irrigation treatments were based on the reference evapotranspiration values, it is necessary that the authors include in detail the procedure for estimating ETo. On the other hand, given that weather conditions change from year to year, considering the average value recorded over the last 5 years for ETo does not seem to be the best strategy for its estimation, considering of climate change.

Lines 134 to 136: “The frequency of irrigation events was every 2 d and soil volumetric water content (%) was monitored every 15 d with a Diviner 2000 portable soil moisture probe (Sentek, Stepney, Australia) for the duration of the experiment”; Since in the work the results obtained depend on the irrigation, the authors must include: 1. How many irrigation events were applied in each treatment, 2. How much water was applied as irrigation in each treatment ; 3. The formula the authors used to estimate soil moisture content; and, 4. Include a graph that shows the time course of soil moisture in each treatment, indicating the events of water application to the crop by irrigation or rain. The latter is important since, as the authors point out, the soil microbial activity depends on its moisture content.

2.3 Microbiological analysis of soil and roots

How many times have the determinations described in this section been made?; One at the end of the experiment?; Several during the experimental work? Soil samples were taken from the experimental units corresponding to each treatment; in all blocks and repetitions?.

2.4 Plant physiological analysis

The measurements described in this section were carried out only once at the end of the harvest?; what date was the harvest done? ; were the measurements made in the plants of the experimental units corresponding to each treatment? ; in all blocks and repetitions?.

2.5 Fruit yield and chemical compounds

Lines 206 and 207: “Fruit productivity (g plant-1) was measured immediately after hand harvesting at 130 d after full flowering [6]”; When??.

The measurements described in this section were carried out on the fruits of the plants of the experimental units corresponding to each treatment?; in all blocks and repetitions?.

3. Results

3.1 Biological properties of soil and roots

Line 244: “The highest FDA activity was reached…”; Authors should include what the abbreviation FDA stands for.

3.2 Plant Physiological Parameters

Line 281: “The maximum quantum yield of photosystem II (Fv/Fm; Figure 4a)…”; Authors should include what the abbreviations Fv and Fm stands for.

Line 302: “The LAI was not influenced by…”; Authors should include what the abbreviation LAI stands for.

Lines 304 and 305: “…significantly increased SPAD…”; Authors should include what the abbreviation SPAD stands for.

3.3 Fruit yield and chemical parameters

Lines 315 to 318: “Fruit yield per plant was significantly higher with 50% ET0 irrigation (p < 0.05) compared to the rest of the treatments, reaching a mean value of 359 g plant-1, while the treatments of 0%, 150% and 100% ET0 replenishment registered mean values of 253, 226 and 193 g plant-1, respectively, with nonsignificant differences among them (Figure 6a)”; How can the authors explain that in the treatment that did not receive irrigation water, the fruit yield was higher than that observed in the 100 and 150% ETo treatments??

Line 322: “…gallic acid 100 g-1 FW, respectively…“; Authors should include what the abbreviation FW stands for.

Lines 324 and 325: “Likewise, treatments with irrigation significantly influenced the antioxidant capacity DPPH (Figure 6c) and ORAC (Figure 6d)”; Authors should include what the abbreviations DPPH and ORAC stands for.

4. Discussion

4.1 Microbiological parameters of soil and roots

Lines 410 and 411: “The lower increases in FDA by irrigation at 150% ET0 indicate that this irrigation dose probably reduces soil aeration and/or porosity”; Porosity is a physical property of soil that is not affected by its moisture content. On the other hand, it is suggested that the authors include the soil moisture content values for the different treatments, in order to check if the 150% ET0 treatment reduces soil aeration. Also, it would be very handy if the authors included the soil porosity value(s).

4.2 Plant physiological parameters

Lines 482 to 484: “Another important finding in the present investigation is that 50% and 100% ET0 treatments positively influenced leaf stomatal conductance, presenting higher values at the beginning of the day”; Why do the authors consider that this behavior was not observed in the 150% ET0 treatment?.

Lines 485 to 487: “Therefore, we believe that in responses to stress conditions (Treatments 0% and 150% ET0) the plant probably closed its stomata preventing gas exchange and water losses”; Excess moisture causes stomata to close?.

Lines 502 and 503: “This can be considered a significant advance in its level of adaptation to drought conditions and excessive soil moisture”; To show that there was excess soil moisture, it is necessary for the authors to include a graph showing the time course of soil moisture content at each treatment, comparing it to porosity. This will make it possible to observe the level of humidity and aeration of the soil at each treatment.

Author Response

Dear Reviewer 2: Thank you very much for your suggestions and comments, they will help improve the quality of the manuscript.

Remark 1: It is however necessary that the authors describe in a better way how the reference evapotranspiration ET0 was calculated, because its value was used as a basis for the definition of the irrigation treatments. 

Response 1: It is included in the manuscript how the reference evapotranspiration ET0 is calculated.

Remark 2: It is also necessary that the authors present in detail the temporal evolution of the soil moisture content in each treatment. This last information is important since, as presented by the authors in the manuscript, almost all the microbiological and physiological variables of the crop depend on the soil moisture content.

Response 2: The temporal evolution of soil moisture content in each treatment was included in the manuscript.

Remark 3: Line 101: “…and accumulated rainfall of 521.1 mm concentrated…”; …and annual accumulated rainfall of 521.1 mm…

Response 3: The modification was included.

Remark 4: Line 108: “The total number of plants in the orchard was 352…”; on what surface?.

Response 4: The surface was included in the manuscript.

Remark 5: Lines 125 to 128: “To determine the actual daily irrigation, ET0 was calculated from the mean values of the previous 5 years taking as a reference what was suggested by Romero et al. [28], with data obtained from the agroclimatic station of INIA Quilamapu located near the study site [29]”; Considering that the irrigation treatments were based on the reference evapotranspiration values, it is necessary that the authors include in detail the procedure for estimating ETo. On the other hand, given that weather conditions change from year to year, considering the average value recorded over the last 5 years for ETo does not seem to be the best strategy for its estimation, considering of climate change.

Response 5: Removed paragraph from the manuscript that mentions “To determine the actual daily irrigation, ET0 was calculated from the mean values of the previous 5 years”. The ET0 estimation procedure was incorporated into the manuscript. At the same time, the ET0 estimation strategy was modified, which was the real annual ET0 estimate.

Remark 6: Lines 134 to 136: “The frequency of irrigation events was every 2 d and soil volumetric water content (%) was monitored every 15 d with a Diviner 2000 portable soil moisture probe (Sentek, Stepney, Australia) for the duration of the experiment”; Since in the work the results obtained depend on the irrigation, the authors must include: 1. How many irrigation events were applied in each treatment, 2. How much water was applied as irrigation in each treatment ; 3. The formula the authors used to estimate soil moisture content; and, 4. Include a graph that shows the time course of soil moisture in each treatment, indicating the events of water application to the crop by irrigation or rain. The latter is important since, as the authors point out, the soil microbial activity depends on its moisture content.

Response 6: What was requested in points 1, 2, 3 and 4 was incorporated. However, in point 4 it was decided to incorporate Table 2, which indicates the moisture content of the soil at different measurement times during the period that the experiment lasted.

Remark 7: How many times have the determinations described in this section been made?; One at the end of the experiment?; Several during the experimental work? Soil samples were taken from the experimental units corresponding to each treatment; in all blocks and repetitions?..

Response 7: Details of the determination of sampling and experimental design of soil and roots were incorporated into the manuscript.

Remark 8: The measurements described in this section were carried out only once at the end of the harvest?; what date was the harvest done? ; were the measurements made in the plants of the experimental units corresponding to each treatment? ; in all blocks and repetitions?.

Response 8: Details of the determination of the sampling and experimental design of physiological measurements of the plant were incorporated in the manuscript.

Remark 9: Lines 206 and 207: “Fruit productivity (g plant-1) was measured immediately after hand harvesting at 130 d after full flowering [6]”; When??..

 Response 9: Added detail on the harvest date in the manuscript.

Remark 10: The measurements described in this section were carried out on the fruits of the plants of the experimental units corresponding to each treatment?; in all blocks and repetitions?.

Response 10: Detail on the determination of the sampling and experimental design of fruit parameters was incorporated in the manuscript.

Remark 11: Line 244: “The highest FDA activity was reached…”; Authors should include what the abbreviation FDA stands for.

Response 11:  The modification was included..

Remark 12: Line 281: “The maximum quantum yield of photosystem II (Fv/Fm; Figure 4a)…”; Authors should include what the abbreviations Fv and Fm stands for..

Response 12:  An abbreviation is written in materials and methods and it is made explicit in the text. Line 281: Fv/Fm corresponds to the maximum quantum yield of photosystem II.

Remark 13: Line 302: “The LAI was not influenced by…”; Authors should include what the abbreviation LAI stands for..

Response 13:  The modification was included...

Remark 14: Lines 304 and 305: “…significantly increased SPAD…”; Authors should include what the abbreviation SPAD stands for..

Response 14:  The modification was included...

Remark 15: Lines 315 to 318: “Fruit yield per plant was significantly higher with 50% ET0 irrigation (p < 0.05) compared to the rest of the treatments, reaching a mean value of 359 g plant-1, while the treatments of 0%, 150% and 100% ET0 replenishment registered mean values of 253, 226 and 193 g plant-1, respectively, with nonsignificant differences among them (Figure 6a)”; How can the authors explain that in the treatment that did not receive irrigation water, the fruit yield was higher than that observed in the 100 and 150% ETo treatments??

Response 15:  In the discussion of the manuscript, the explanation based on previous studies of calafate, chapter 4.3 “Fruit yield and chemical parameters”.

Remark 16: Line 322: “…gallic acid 100 g-1 FW, respectively…“; Authors should include what the abbreviation FW stands for..

Response 16:  The modification was included.

Remark 17: Lines 324 and 325: “Likewise, treatments with irrigation significantly influenced the antioxidant capacity DPPH (Figure 6c) and ORAC (Figure 6d)”; Authors should include what the abbreviations DPPH and ORAC stands for.

Response 17:  The modification was included.

Remark 18: Lines 410 and 411: “The lower increases in FDA by irrigation at 150% ET0 indicate that this irrigation dose probably reduces soil aeration and/or porosity”; Porosity is a physical property of soil that is not affected by its moisture content. On the other hand, it is suggested that the authors include the soil moisture content values for the different treatments, in order to check if the 150% ET0 treatment reduces soil aeration. Also, it would be very handy if the authors included the soil porosity value(s)..

Response 18:   We agree with what was stated by the reviewer. Irrigation dose does not affect soil porosity, therefore this paragraph was modified in the manuscript. Table 2 was incorporated, which indicates the moisture content of the soil at different measurement times during the period in which the test lasted. In discussion, the porosity value of the soil of the study site was included in chapter 4.1 "Microbiological parameters of soil and roots".

Remark 19: Lines 482 to 484: “Another important finding in the present investigation is that 50% and 100% ET0 treatments positively influenced leaf stomatal conductance, presenting higher values at the beginning of the day”; Why do the authors consider that this behavior was not observed in the 150% ET0 treatment?.

Response 19 In the discussion of the manuscript, the explanation based on previous studies where irrigation doses over 100% of the ET0 effect stomatal conductance in blueberries is incorporated, chapter 4.3 "Fruit yield and chemical parameters". Chapter 4.2 “Plant physiological parameters”.

Remark 20: Lines 485 to 487: “Therefore, we believe that in responses to stress conditions (Treatments 0% and 150% ET0) the plant probably closed its stomata preventing gas exchange and water losses”; Excess moisture causes stomata to close?.

Response 20:  In accordance with the above, excess humidity decreases gas exchange and could even induce stomatal closure in species such as blueberries. This paragraph was modified in the manuscript.

Remark 21: Lines 502 and 503: “This can be considered a significant advance in its level of adaptation to drought conditions and excessive soil moisture”; To show that there was excess soil moisture, it is necessary for the authors to include a graph showing the time course of soil moisture content at each treatment, comparing it to porosity. This will make it possible to observe the level of humidity and aeration of the soil at each treatment..

Response 21:   Table 2 was included in the manuscript, indicating the soil moisture content on different dates throughout the experiment. In the manuscript mention is also made of the porosity of the soil of the study site. 

Reviewer 3 Report

Dear authors,

I enjoyed reading your manuscript and found it interesting and promising. It falls well into the scope of the special issue of Horticulturae it was submitted to. Please consider all the observations below when preparing your revised version:

Line 67: preposition missing: allow TO set up (or allow set up OF)

Line 82: fruit-PRODUCING (or fruit-BEARING) species

Line 85: I believe you meant to say water quantities (not qualities)

Lines 138-139: "being" should be moved before "averaged"

Line 147: rephrase to "was measured using a spectrophotometer"

Line 159: more details are needed here. How was the ammonium chloride standard curve prepared? What is the exact analytical property measured? What is the equation that links this analytical signal to the concentration of NH4+?

Lines 165 and 212: Centrifuge speed should better be expressed in terms of g force rather than revolutions per minute, as the latter are not the same for all centrifuges

Lines 166, 173, 213, 220: Please provide the standard curve equations used, including the linear ranges for their validity.

Line 211: Mention type/make of the ultrasonic bath used.

Line 231: "aforementioned" (one word)

Figure 4(a): Although color-coded, there is still a lot of overlap of data points and error bars, making the chart difficult to comprehend, so I recommend turning it also into a bar chart just like in figure 4(b)

Table 3: I do not understand why are the results presented as equivalent mg of cyanidin-3-glucoside per 100 g" if each different anthocyanin was successfully determined individually after HPLC separation. Please clarify! HPLC is a technique that allows individual quantification of compounds unlike bulk spectrophotometric measurements.

Line 628-629: rephrase to "soil microbiological evaluations should not be neglected"

Throughout the manuscript: "d" is not the standard abbreviation for days; either replace it with the full word "days" or define it as an abbreviation

Also, some acronyms are not defined at the time of their first occurrence in the text, e.g. DPPH, FW, ORAC, HPLC-DAD, LSD etc. This should be addressed! Careful also that FW is defined somewhere as "first watering", but then the same acronym is used to mean (presumably) "fresh weight"

Author Response

Dear Reviewer 3: Thank you very much for your suggestions and comments, they will help improve the quality of the manuscript.

Remark 1: Line 67: preposition missing: allow TO set up (or allow set up OF).

Response 1:   Paragraph modified.

Remark 2: Line 82: fruit-PRODUCING (or fruit-BEARING) species

Response 2:  Paragraph modified.

Remark 3: Line 85: I believe you meant to say water quantities (not qualities)

Response 3:  Paragraph modified.

Remark 4: Lines 138-139: "being" should be moved before "averaged"

Response 4:   Paragraph modified.

Remark 5: Line 147: rephrase to "was measured using a spectrophotometer"

Response 5:   Paragraph modified

Remark 6: Line 159: more details are needed here. How was the ammonium chloride standard curve prepared? What is the exact analytical property measured? What is the equation that links this analytical signal to the concentration of NH4+?

Response 6:   Methodology for estimating urease activity in manuscript was modified.

Remark 7: Lines 165 and 212: Centrifuge speed should better be expressed in terms of g force rather than revolutions per minute, as the latter are not the same for all centrifuges

Response 7:   We agree with the above, the modification was included.

Remark 8: Lines 166, 173, 213, 220: Please provide the standard curve equations used, including the linear ranges for their validity. Line 211: Mention type/make of the ultrasonic bath used.

Response 8 The modification was included in lines 166, 173, 213, 220, and line 211 .

Remark 9: Line 231: "aforementioned" (one word)

Response 9:   The modification was included.

Remark 10: Figure 4(a): Although color-coded, there is still a lot of overlap of data points and error bars, making the chart difficult to comprehend, so I recommend turning it also into a bar chart just like in figure 4(b)

Response 10:    Regarding the above, we believe that the bar graph does not allow a better visualization of the trend of the maximum efficiency of chlorophyll, but not the line graph. We have modified the published graphics. There are scientific publications that use the same format. However, an adjustment was made to the size of the axes to improve the display.

Remark 11: Table 3: I do not understand why are the results presented as equivalent mg of cyanidin-3-glucoside per 100 g" if each different anthocyanin was successfully determined individually after HPLC separation. Please clarify! HPLC is a technique that allows individual quantification of compounds unlike bulk spectrophotometric measurements.

Response 11:    It is a writing error, it is expressed as mg 100 g-1FW of sample. They were quantified with a cyanidin-3-glucoside curve. But, although they are quantified with cyanidin-3-glucoside, the equivalent is calculated for each of the identified anthocyanins. The modifications were included in the manuscript..

Remark 12: Line 628-629: rephrase to "soil microbiological evaluations should not be neglected"

Response 12:    The modifications were included in the manuscript.

Remark 13: Throughout the manuscript: "d" is not the standard abbreviation for days; either replace it with the full word "days" or define it as an abbreviation

Response 13:    The modifications were included in the manuscript.

Remark 14: Also, some acronyms are not defined at the time of their first occurrence in the text, e.g. DPPH, FW, ORAC, HPLC-DAD, LSD etc. This should be addressed! Careful also that FW is defined somewhere as "first watering", but then the same acronym is used to mean (presumably) "fresh weight"

Response 14:    The modifications were included in the manuscript.
